# Structural and electronic determinants of lytic polysaccharide monooxygenase reactivity on polysaccharide substrates

T.J. Simmons[1], K.E.H. Frandsen [9], L. Ciano [3], T. Tryfona[1], N. Lenfant[4,5], J.C. Poulsen[2], L.F.L. Wilson [1], T. Tandrup[2], M. Tovborg[6], K. Schnorr[6], K.S. Johansen[7], B. Henrissat[4,5,8], P.H. Walton [3], L Lo Leggio [2] & P. Dupree [1]

Lytic polysaccharide monooxygenases (LPMOs) are industrially important copper-dependent enzymes that oxidatively cleave polysaccharides. Here we present a functional and structural characterization of two closely related AA9-family LPMOs from *Lentinus similis* (*Ls*AA9A) and *Collariella virescens* (*Cv*AA9A). *Ls*AA9A and *Cv*AA9A cleave a range of polysaccharides, including cellulose, xyloglucan, mixed-linkage glucan and glucomannan. *Ls*AA9A additionally cleaves isolated xylan substrates. The structures of *Cv*AA9A and of *Ls*AA9A bound to cel-lulosic and non-cellulosic oligosaccharides provide insight into the molecular determinants of their specificity. Spectroscopic measurements reveal differences in copper co-ordination upon the binding of xylan and glucans. *Ls*AA9A activity is less sensitive to the reducing agent potential when cleaving xylan, suggesting that distinct catalytic mechanisms exist for xylan and glucan cleavage. Overall, these data show that AA9 LPMOs can display different apparent substrate specificities dependent upon both productive protein–carbohydrate interactions across a binding surface and also electronic considerations at the copper active site.

[1] Department of Biochemistry, University of Cambridge, Cambridge, CB2 1QW, UK. [2] Department of Chemistry, University of Copenhagen, Copenhagen, DK-2100, Denmark. [3] Department of Chemistry, University of York, York, YO10 5DD, UK. [4] Architecture et Fonction des Macromolécules Biologiques (AFMB), CNRS, Aix-Marseille Université, Marseille, 13288, France. [5] Institut National de la Recherche Agronomique (INRA), AFMB, Marseille, 13288, France. [6] Research and Development, Novozymes A/S, Bagsvaerd, DK-2880, Denmark. [7] Department of Geoscience and Natural Resources Management, Copenhagen University, Frederiksberg, DK-1958, Denmark. [8] Department of Biological Sciences, King Abdulaziz University, Jeddah, 21589, Saudi Arabia. [9] INRA, UMR 1163 BBF (Biodiversité et Biotechnologie Fongiques), Marseille, 13288, France. Simmons T.J. and Frandsen K.E.H. contributed equally to this work. Correspondence and requests for materials should be addressed to L.L.L. (email: leila@chem.ku.dk) or to P.D. (email: pd101@cam.ac.uk)

The need for sustainable sources of energy and materials has spurred significant research efforts towards a greater understanding of the biological catabolism of lignocellulose, the world's most abundant source of renewable material and bioenergy[1, 2]. The inherent recalcitrance of lignocellulose, however, is one of the major barriers to the utilization of biomass. This recalcitrance is a consequence of both the heterogeneous composition and the often semi-crystalline association of the polymers[3, 4]. In addressing the problem of recalcitrance, multiple potential means have been proposed and assessed, including chemical, mechanical and enzymatic methods. Advances in enzyme cocktail formulations that accelerate the saccharification step of cell wall breakdown[5], in particular the inclusion of the lytic polysaccharide mono-oxygenases (LPMOs)[6], are helping cellulosic-ethanol biorefineries move towards both commercial and environmental viability.

LPMOs are reducing agent- and $O_2$-dependent copper metalloenzymes now classified as auxiliary activity families AA9–AA11 and AA13[7–11]. Extensive spectroscopic and structural studies on LPMOs have shown that the enzyme's active site contains a single copper ion, which is coordinated by the amino terminus nitrogen atom, by a side chain nitrogen atom of the N-terminal histidine, and by the side chain nitrogen atom of an additional histidine, in a structural motif known as the histidine brace[12]. What is distinctive about LPMOs is that they oxidatively rather than hydrolytically cleave polysaccharides producing saccharides with oxidized ends[13]. LPMOs augment the action of other polysaccharide-degrading enzymes, and accordingly much research attention is devoted to a greater understanding of the enzymatic mechanism and the range of LPMO saccharide substrates.

It was first shown that LPMOs could boost the action of cellulases on cellulose and chitin[14–16], but LPMOs are now known to act on several crystalline substrates such as chitin, cellulose and retrograded starch[10–12, 17]. Later, enzymes with activity against non-crystalline and oligomeric structures were identified[18, 19]. Furthermore, fungal AA9 LPMOs have been shown to be active on soluble substrates such as xyloglucan, mixed-linkage glucan and glucomannan[19–22], and on cellulose-bound xylan[23]. Conspicuously, an LPMO active on isolated xylan has not been reported; however, this range of reported substrates will likely grow. The large number and sequence diversity of LPMOs that individual fungi maintain[24], and their disparate expression profiles when the fungi are grown on different polysaccharide substrates[25, 26], signal that AA9 LPMOs do have distinct, and functionally significant, polysaccharide substrate specificities, although some evolutionary diversity of LPMOs likely arises through their use of different reducing systems[27, 28].

The root causes of LPMO substrate specificity remain poorly understood. This is because LPMO chemistry is a subtle and complex combination of structural and electronic factors, both of which must be taken into account when developing an understanding of the mechanism of action[29]. The structure–function relationship of substrate specificity and regiospecificity has been recently reviewed[30, 31]. Insight into LPMO:substrate binding can be gained from the structures of LPMOs[30] and combined structural and spectroscopic studies of LPMOs in contact with substrate. Recent ITC, NMR and docking studies of an AA9 LPMO from *Neurospora crassa* in contact with oligosaccharides revealed that more extended substrates had significantly higher binding affinities. This is in accord with a multi-point interaction of the substrate with the LPMO surface where the surface loops in some LPMOs remote from the active site enhance binding affinity[21]. The study also showed that a single cellohexaose (Cell$_6$) chain likely spans the copper active site from the −3 to +3 or −2 to +4 subsites (subdivisions of the binding cleft numbered relative to the site of cleavage[32]), in which the L3 loop (important for interactions with the +3/+4 subsites) and the LC loop (important for binding to approximately −4 subsite) lie at somewhat extended distances from the copper active site. Detailed insight into an AA9 LPMO–substrate interaction came from the first crystal structures of LPMO:oligosaccharide complexes: *Lentinus similis* AA9A (*Ls*AA9A) bound to Cell$_6$ and Cell$_3$[33]. Cell$_6$ was shown to bind at subsites −4 to +2 via interactions with aromatic residues, the N-terminal His and a conserved Tyr as well as a number of hydrogen-bonds with other residues in a contoured binding surface on the LPMO. The +2 glucosyl residue exhibits a set of well-defined hydrogen-bonding interactions with amino-acid side chains (Asn28, His66 and Asn67) essentially locking this residue into a fixed position with respect to the active site.

Electronic factors around the active site also play a key role in determining the reactive mechanism. Changes in the electronic structure of the copper ion, an important factor in the ability of the copper ion to activate $O_2$, occur upon substrate binding to *Ls*AA9A[33]. Furthermore, in an illustration of the complexity of substrate–LPMO interaction and the subtle interplay of electronic and structural factors, Cell$_6$ is bound synergistically with an exogenous ligand on the copper ion. It is likely that the oxidative mechanism adopted by LPMOs can proceed via one or more of several different routes[34], the determinants of which depend on the varying extents of the substrate, the reducing agent reducing the potential and the positioning of the substrate on the LPMO surface. For instance, the means by which electrons are donated to the LPMO active site modulate the apparent range of reactivity[28, 35].

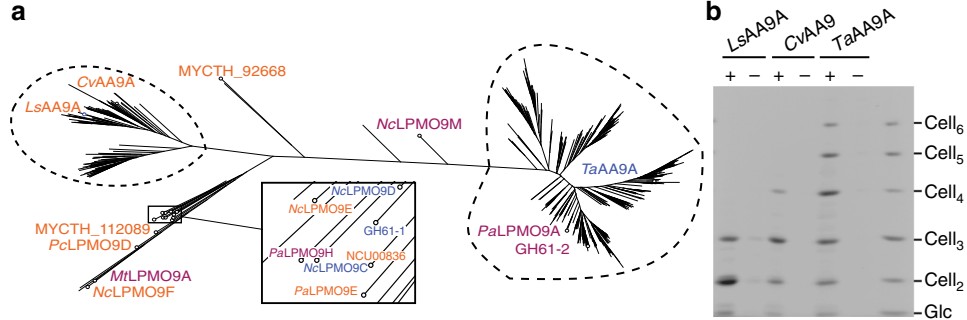

**Fig. 1** Sequence similarity between *Ls*AA9A, *Cv*AA9A and *Ta*AA9A, and analysis of their reaction products. **a** Distance tree of 444 selected AA9 sequences (see Methods). Blue, purple and orange labels designate AA9 enzymes that oxidize the sugar ring at C1, C4 and C1+C4, respectively. See Supplementary Table 1 for protein accession numbers. Unlabelled branches represent AA9 enzymes for which the regioselectivity of oxidation is not available from the literature. **b** PACE gel showing reaction products of the three enzymes on PASC; +, incubation with 4 mM ascorbate; −, incubation without ascorbate (performed in triplicate)

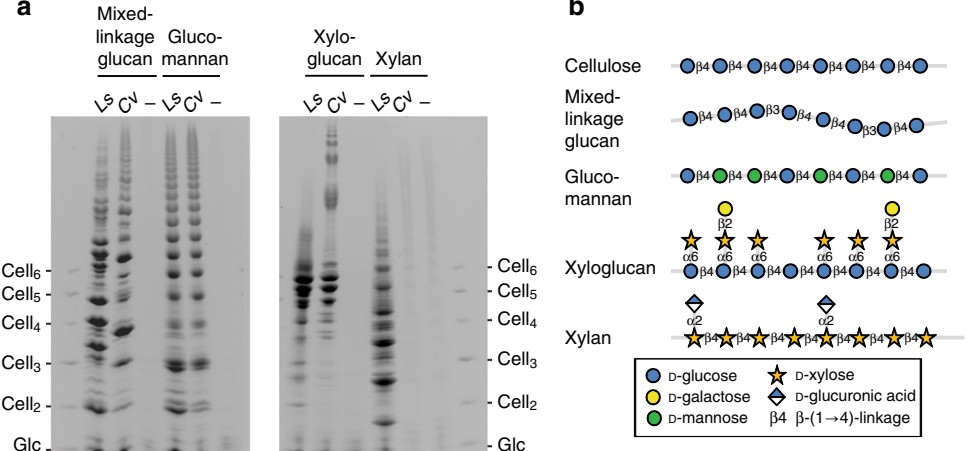

**Fig. 2** Comparison of *Ls*AA9A and *Cv*AA9A action on non-cellulosic substrates. **a** PACE gel showing digestion products on lignocellulosic polysaccharides with 4 mM ascorbate reducing agent. *Ls*, *Ls*AA9A; *Cv*, *Cv*AA9A; −, no enzyme (performed in triplicate). The migration standards are cello-oligosaccharides. **b** Structures of polysaccharides

The detailed molecular and electronic insights of the LPMO–substrate interaction afforded by combined biochemical, X-ray diffraction and electron paramagnetic resonance (EPR) spectroscopic studies can significantly enhance our understanding of LPMO reactivity. We report herein a study into the principal structural and electronic factors of the reactivity of two AA9 LPMOs with a range of substrates. Through X-ray crystal structures studies of *Ls*AA9A with bound substrates, we illustrate how binding cleft interactions dictate the site of polysaccharide attack. Through comparison with the *Cv*AA9A structure, also determined here, we suggest some structural determinants of specificity for the two enzymes. *Ls*AA9A is active on isolated xylan, but this activity is associated with a distinct low sensitivity to reducing agent potential and a different copper co-ordination at the active site, which together reveal an alternative mechanistic pathway for LPMO action on this substrate. These data show how AA9 LPMO substrate cleavage is dependent upon both productive protein:carbohydrate interactions across a binding surface and also electronic considerations at the active site.

## Results

***Cv*AA9A is an AA9 LPMO active on cello-oligosaccharides**. To help understand the basis of AA9 substrate specificity, we searched for enzymes related to *Ls*AA9A that might also cleave soluble oligosaccharides. Because LPMOs exhibit high variability in their C-termini, we performed a large-scale alignment of LPMO protein sequences using their N-terminal portion[36]. We selected 98 AA9 sequences that were highly similar in their N-terminal half to *Ls*AA9A and 326 sequences that were highly similar to *Ta*AA9A in the same region. *Ta*AA9A was used for comparison purposes, as it does not show the ability to cleave soluble oligosaccharides. After adding the sequences of 20 AA9 enzymes studied in the literature, a distance tree was built with the resulting 444 AA9 sequences (Fig. 1a, Supplementary Table 1). The tree clearly places *Ls*AA9A and *Ta*AA9A in distinct clades. From the *Ls*AA9A clade, we identified an LPMO from *Collariella virescens* (*Cv*AA9A; 46% sequence similarity to *Ls*AA9A) that lacks some residues observed by Frandsen et al.[33] as being involved in enzyme–substrate interactions (Supplementary Fig. 1). All three of the subsite +2 substrate-binding residues in *Ls*AA9A (Asn28, His66 and Asn67) are different in *Cv*AA9A (Thr28, Arg67 and Val68) (Supplementary Fig. 1). To study the activity of *Cv*AA9A, the enzyme was expressed in *Aspergillus oryzae* and successfully purified from the fermentation broth.

(Note that the expression in this fungal host preserves the natural side-chain methylation at the N-terminal histidine, in contrast to fungal LPMO expression in *Pichia pastoris* and bacterial systems.) On phosphoric acid-swollen cellulose (PASC), *Cv*AA9A produced a range of cello-oligosaccharides (Fig. 1b). The cello-oligosaccharide product profile of *Cv*AA9A was similar to that of *Ls*AA9A and notably shorter than those produced by *Ta*AA9A. Indeed, *Cv*AA9A readily degraded Cell$_6$-(2-aminobenzamide) (Cell$_6$-2AB) using a C4-oxidising mechanism to yield Cell$_3$ and oxidized Cell$_3$-2AB (Supplementary Fig. 2), like *Ls*AA9A but unlike *Ta*AA9A[33]. Therefore, the distance relationships between the three enzymes, as measured using the N-terminal comparison method above, mirror the similarities in activities of the enzymes.

**Position-specific cleavage of a range of hemicelluloses**. We next determined whether the *Ls*AA9A and *Cv*AA9A enzymes are active on a range of β-(1 → 4)-D-glucan-related polysaccharides (Fig. 2, Table 1). Mixed-linkage glucan (MLG) is a β-D-glucan in which three to four (1 → 4)-linked residues (Cell$_3$, Cell$_4$) are separated by single (1 → 3) bonds, glucomannan has a backbone randomly composed of β-(1 → 4)-D-glucosyl and β-(1 → 4)-D-mannosyl residues, xyloglucan is a β-(1 → 4)-D-glucan with α-(1 → 6)-D-xylosyl branches, and xylan is a polymer of β-(1 → 4)-D-xylosyl residues that is similar to β-(1 → 4)-D-glucan but lacks C6 groups (Fig. 2b). Both *Ls*AA9A and *Cv*AA9A showed activity against MLG, glucomannan and xyloglucan, producing a range of oligosaccharide products (Fig. 2a). *Ls*AA9A also showed some activity on xylan, whereas *Cv*AA9A showed no measurable activity on this substrate. No LPMO activity was observed on starch (α-(1 → 4)-D-glucan), laminarin (β-(1 → 3)-D-glucan) or chitin (poly β-(1 → 4)-D-GlcNAc) (Supplementary Fig. 3). Altogether, these activities indicate that both *Ls*AA9A and *Cv*AA9A enzymes only cleave near β-(1 → 4) bonds, and that some variation to the cellulosic β-(1 → 4)-D-glucan, including substitution, linkage and backbone residue, can be accommodated at or near the site of cleavage by both of the enzymes.

To identify precise substrate cleavage sites, we studied the products of both *Ls*AA9A and *Cv*AA9A cleavage of MLG, glucomannan, xyloglucan and xylan (in the case of *Ls*AA9A) polysaccharides by MALDI-ToF MS. Minor double oxidation products were observed, indicating cleavage of these hemicelluloses and PASC using both C1- and C4-oxidising mechanisms (Fig. 3; Supplementary Fig. 4, Table 1). We further investigated the site of attack on these different hemicelluloses using differing

**Table 1 Summary of activity assays on different substrates**

| Polysaccharide | LsAA9A | | CvAA9A | |
|---|---|---|---|---|
| | Activity | Notes | Activity | Notes |
| Cellulose and cello-oligosaccharides | ++ | Activity on both cellulose oligosaccharides and insoluble cellulose material (PASC) | ++ | Activity on both cellulose oligosaccharides and insoluble cellulose material (PASC) |
| MLG | ++ | Pattern suggests β-(1 → 3) bonds accommodated at specific places within active site, but not between −1 and +1 | ++ | Pattern suggests β-(1 → 3) bonds accommodated at specific places within active site, but not between −1 and +1 |
| Glucomannan and Man₆ | ++ | Cleavage can occur with Glc or mannose at −1 or +1. Inactivity on Man₆ indicates some Glc C2 hydroxyl orientation needs to be present between −3 and +3 | ++ | Cleavage can occur with Glc or mannose at −1 or +1. Inactivity on Man₆ indicates some Glc C2 hydroxyl orientation needs to be present between −3 and +3 |
| Xyloglucan | ++ | Cleavage occurs with unsubstituted Glc at subsite +1. Xylosyl substitution at −3, −2, −1, +2 and +3. Galactosyl-xylosyl substitutions can occur at either −2, −1 and/or +3 | ++ | Cleavage occurs with unsubstituted Glc at subsite +1. Xylosyl substitution at −3, −2, −1, +2 and +3. Galactosyl-xylosyl substitutions can occur at either −2, −1 and/or +3 |
| Xylan and Xyl₆ | + | Activity on both. Weak Xyl₆ activity compared to Cell₆ suggests that, while Glc C6 is not required for activity, it is very important at certain sites, such as +1 | +/− | Much poorer activity of CvAA9A compared with LsAA9A |
| Starch | − | Absence of activity indicates that LsAA9A necessarily cleaves β-(1 → 4) bonds | − | Absence of activity indicates that CvAA9A necessarily cleaves β-(1 → 4) bonds |
| Laminarin | − | | − | |
| G4G3G4G (MLG oligosaccharide) | − | | − | |
| Chitin | − | Absence of activity indicates that LsAA9A either requires O2 interactions or cannot accommodate N-acetyl on amino C2 | − | Absence of activity indicates that CvAA9A either requires O2 interactions or cannot accommodate N-acetyl on amino C2 |

Semi-quantitative activity results summarising the activity of LsAA9A and CvAA9A on the range of different substrates used in this manuscript. All experiments were carried out in at least triplicate

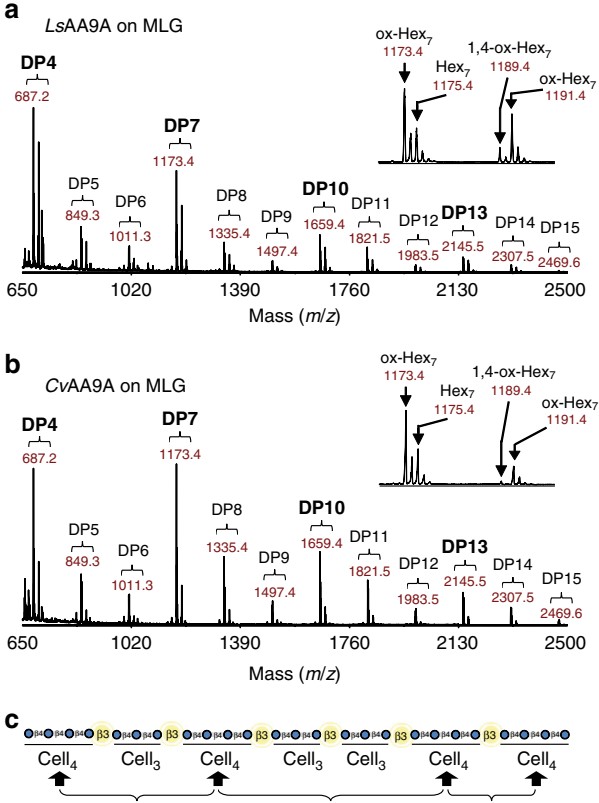

**Fig. 3** LsAA9A and CvAA9A digestion products of MLG suggest preference for Cell₄ region cleavage. Products of LsAA9A (**a**) and CvAA9A (**b**) activity on barley MLG with 4 mM ascorbate were analysed by MALDI-ToF MS (performed in triplicate). Both enzymes can produce both C1 and C4 oxidation on MLG (1,4-ox; oxidized C1 and C4, see insets). Further, oligosaccharide profiles show a distinct pattern indicative of the mechanism of attack and substrate specificity of each enzyme on MLG. **c** Proposed region of cleavage

protocols. On MLG, we observed in the MALDI data a predominance of DP 4, 7 and 10 oligosaccharides indicating that each enzyme favours cleaving within Cell₄ regions over Cell₃ regions (Fig. 3). The inability of LsAA9A and CvAA9A to cleave β-Glc-(1 → 4)-β-Glc-(1 → 3)-β-Glc-(1 → 4)-Glc (G4G3G4G), despite their ability to cleave Cell₄ (G4G4G4G) (Supplementary

Fig. 5), supports the hypothesis that neither enzyme can cleave at β-(1 → 3) bonds and require substantial β-(1 → 4)-linked regions for cleavage. On glucomannan, we employed High-Performance Anion-Exchange Chromatography (HPAEC) analysis of trifluoroacetic acid (TFA) hydrolysates of digestion products to assess the site of cleavage. Notably, the data indicated that cleavage can occur not only between glucosyl residues, but also with mannose at the +1 or −1 subsite (Supplementary Fig. 6). In order to deduce site of attack on xyloglucan we employed xyloglucan DP14–18 oligosaccharides (Supplementary Fig. 7). Inspection of the position of substituted glucose (Glc) in the products indicated that xylosyl substitution of Glc at O-6 was accommodated at the −3, −2, −1, +2 and +3 subsites but unsubstituted Glc was always required at subsite +1. In contrast to the LsAA9A and CvAA9A products on polysaccharides, LsAA9A degraded Xyl₆-2AB to yield two trimers using solely a C4-oxidising mechanism (Supplementary Fig. 8), analogous to cleavage of Cell₆-2AB by both LsAA9A[33] and CvAA9A (Supplementary Fig. 2).

To allow a semi-quantitative determination of the influence of sugar structures on enzyme activity, we probed LsAA9A and CvAA9A cleavage of the soluble Cell₆, xylohexaose (Xyl₆) and mannohexaose (Man₆) oligosaccharides (Supplementary Fig. 9). The LsAA9A activity against Cell₆ was substantially (~100-fold) better than its activity on Xyl₆. Consistent with the absence of activity on glucuronoxylan, CvAA9A activity on Xyl₆ was almost undetectable (~1000-fold less than Cell₆ activity). Although both enzymes showed activity on glucomannan and can cleave adjacent to mannose, activity was scarcely detectable on Man₆ (~10,000-fold less than Cell₆), indicating that the enzymes require some Glc residues within a mannan backbone for activity.

Recent results show dependence of the LPMO action on reductant strength[28, 37]. We found that cleavages of MLG, glucomannan and xyloglucan by LsAA9A were sensitive to reducing agent potential, with ascorbate as reductant yielding much higher amount of product (Fig. 4a). In contrast, cleavage of xylan was not sensitive. We corroborated this finding with oligosaccharides, observing that Xyl₆ was poorly sensitive to reducing agent potential, unlike cleavage of Cell₆ where LsAA9A showed much greater activity with ascorbate than pyrogallol[33] (Fig. 4b).

**LsAA9A: and CvAA9A:cello-oligosaccharide structures**. To help understand the structural basis of LPMO attack on different substrates, we employed crystallographic analyses. We report here an LsAA9A:Cell₅ complex (Fig. 5a, b; Supplementary Table 4; Table 2), which, owing to a lack of significant substrate contacts

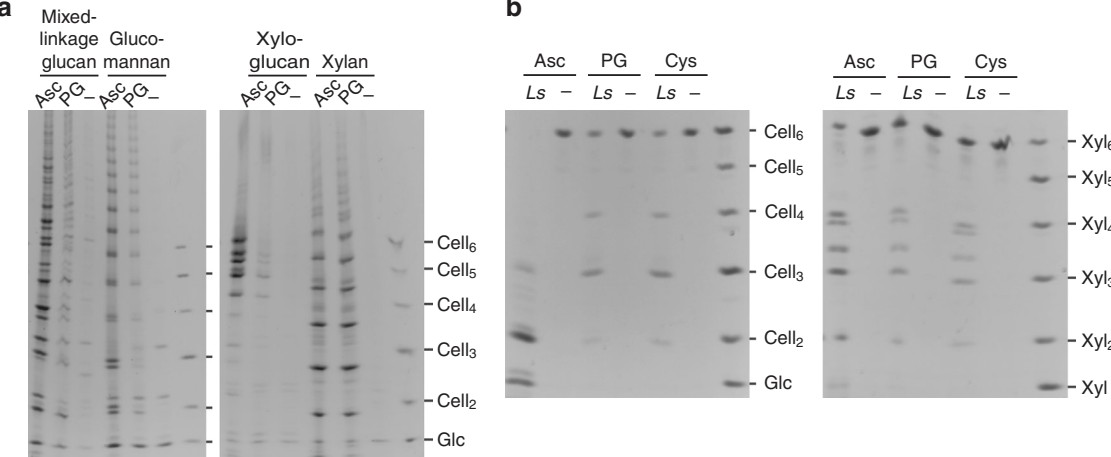

**Fig. 4** *Ls*AA9A activity on different poly- and oligosaccharide substrates show differing sensitivity to reducing agent potential. **a** PACE gels showing products of *Ls*AA9A activity on MLG, glucomannan, xyloglucan and xylan polysaccharides using 4 mM ascorbate or 4 mM pyrogallol as reductants (performed in triplicate). The migration standards are cello-oligosaccharides. **b** PACE gels showing products of *Ls*AA9A activity on Cell₆ and Xyl₆ oligosaccharides using 4 mM ascorbate, pyrogallol and cysteine as reductants (performed in triplicate). Asc Ascorbate; Cys cysteine; PG pyrogallol. The migration standards are cello- and xylo-oligosaccharides

to symmetry-related molecules, is a more faithful depiction of the binding conformation of a single oligosaccharide to *Ls*AA9A as compared to the original *Ls*AA9A:Cell₆ structure described by Frandsen et al.[33]. Tyr203 stacking is still a major interaction in *Ls*AA9A:Cell₅ but a new hydrogen bond is seen between O6 and Asp150 at subsite −3, and glycosidic torsion angles are closer to ideal values (Supplementary Table 2). Other interacting residues at the negative subsites are Glu148, Arg159 and Ser77 (Fig. 5; Supplementary Table 3). Like the *Ls*AA9A:Cell₆ structure[33], the main interactions to Cell₅ are a network of hydrogen bonds by Asn28, His66 and Asn67 interacting with O2 and O3 at subsite +2, and the interaction with MeHis1 at subsite +1[38].

To understand better how protein structure might influence the similarities and differences in *Cv*AA9A and *Ls*AA9A substrate cleavage patterns, the X-ray crystal structure of *Cv*AA9A was solved (Supplementary Figs. 10 and 11; Supplementary Tables 3 and 4; Table 2). The Cu-coordinating amino-acid residues are MeHis1 and His79 (with equatorial distances to the Cu ranging from 2.0 to 2.1 Å), while a non-coordinating Tyr169 occupies the axial position (2.6–2.8 Å). No exogenous ligands are evident within 3.0 Å of the Cu-ion indicating that the active site is mostly in a photoreduced Cu(I) state. A 'pocket-water' is bound in an H-bond network with the amide-nitrogen and oxygen of Asp76 and MeHis1, respectively. The active-site geometry of *Cv*AA9A thus closely resembles that of *Ls*AA9A (Supplementary Fig. 10c). However, there are some amino-acid differences in *Cv*AA9A compared to *Ls*AA9A at subsites +2 and −1. Crystals of *Cv*AA9A were soaked with Cell₃ and Cell₆ oligosaccharides but this did not result in any catalytically relevant complex.

**_Ls_AA9A:hemicellulose oligosaccharide structures**. To study the structural determinants of the *Ls*AA9A positional specificity of cleavage, a number of *Ls*AA9A crystal structures in complex with MLG, glucomannan and xylo-oligosaccharides were solved (see Table 2, Supplementary Fig. 12 and Supplementary Tables 2 and 4 for experimental and crystallographic data and refinement information, hydrogen-bonding interactions between enzyme and ligand, and ligand conformations). Soaking experiments with commercially available xyloglucan fragments failed to produce crystallographic complexes, possibly because the substrate oligosaccharides are large and binding likely to be impeded by crystal contacts.

*Complexes with MLG tetrasaccharide*. *Ls*AA9A crystals were soaked with two different MLG tetrasaccharides, each with a single β-(1 → 3) linkage: G4G4G3G and G4G3G4G. Interestingly, the *Ls*AA9A:G4G4G3G complex did not reveal any β-(1 → 3) linkages. An apparent Cell₄ substrate appears to be bound from subsite −2 to +2 (Supplementary Fig. 13) giving essentially identical interactions as the −2 to +2 glucosyl residues in the *Ls*AA9A:Cell₅ complex. We interpret this result as the β-(1 → 4)-glucan (Cell₃) part of the substrate being bound in two overlapping conformations in different asymmetric units from subsites −2 to +1 and −1 to +2, while the β-(1 → 3)-glucosidic residues are completely disordered in both cases. A structure of *Ls*AA9A crystals soaked with G4G3G4G could not be convincingly modelled, further indicating that the enzyme needs at least two consecutive β-(1 → 4) linkages (a Cell₃ unit) for recognition and efficient binding.

*Complexes with glucomannan oligosaccharides*. *Ls*AA9A crystals were soaked with a mixture of glucomannan oligosaccharides. The resulting difference density was well defined clearly showing glycosyl units occupying subsites −3 to +2, additional density at −4 and some residual density occupying subsite +3 (Fig. 5c, e). Consistent with the activity data, the structure unequivocally showed a mannosyl unit at the +1 subsite, while glucosyl units were clearly observable at −2, −1 and +2 subsites. Moreover the C2 hydroxyl of the mannosyl unit at subsite +1 points towards the face of the imidazole side chain of MeHis1, and the axial water molecule is displaced (Fig. 5d, f). The identity of the glycosyl unit at subsite −3 is ambiguous though best modelled as mannose. The density of the glycosyl unit at subsite −4 is weak and occupies a very similar position as the corresponding unit in the *Ls*AA9A:Cell₆ complex, as does the glycosyl unit at subsite −3, due to similar crystal constraints.

*Complexes with xylo-oligosaccharide*. Although in crystals soaked with Xyl₃ and Xyl₄ the oligosaccharides did not fully span the active site, *Ls*AA9A:Xyl₅ crystals revealed very well-defined density from subsites −3 to +2 (Fig. 6a). The oligosaccharide position at subsites −3 to −1 are similar to *Ls*AA9A:Cell₅, but with a translation of about half a pyranose unit in the non-reducing end direction. In contrast, the plane of the xyloside unit at subsite +1 is rotated ~90° compared to the corresponding glucosidic unit, while the xyloside residue binding +2 is rotated ~180° (Fig. 6d; Supplementary Table 2). As a result

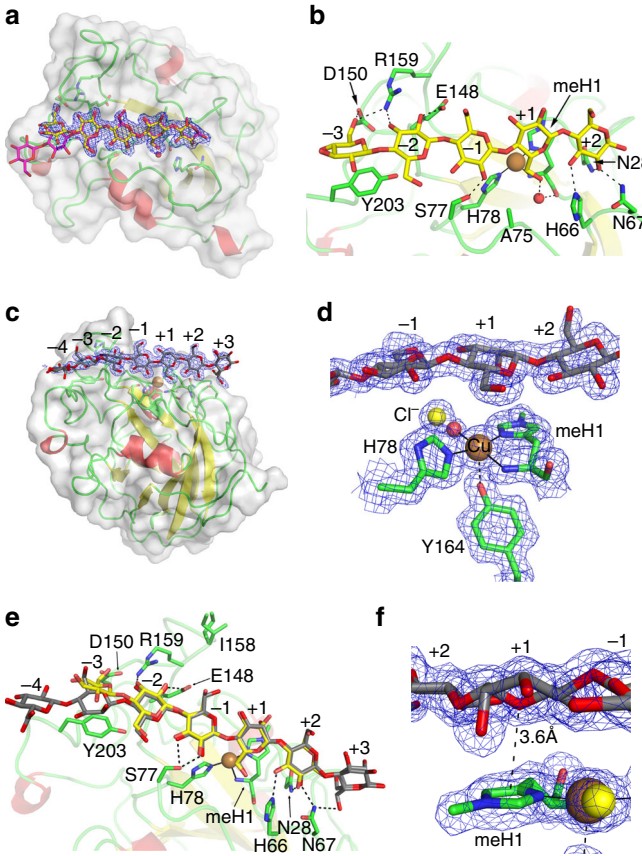

**Fig. 5** Structures of *Ls*AA9A:Cell₅ and *Ls*AA9A:glucomannan oligosaccharide complexes. **a** Cell₅ (yellow) is well defined in subsites –3 to +2. A 2$F_{obs}$–$F_{calc}$ electron density map is shown at 1σ contour level. The structure shows no crystal contact induced distortion of the Cell₅ substrate when compared to Cell₆ (magenta). **b**, *Ls*AA9A-Cell₅ interactions are shown as dashes. An additional interaction between subsite –3 O6 and Asp150 is gained in the absence of symmetry related contacts to the substrate. **c** Overall structure of *Ls*AA9A with GM (grey) bound from subsite −4 to +3. **d** Close-up (subsite –1 to +2) of the active site with GM fragment bound. Axial coordinations are in black dashes while equatorial coordinations are in full black lines. **e** Top-down view of *Ls*AA9A:GM (GM in grey) and for comparison *Ls*AA9A:Cell₅ (Cell₅ in yellow). Dashed lines show interactions within hydrogen bond distance (2.8 Å). **f** The C2-hydroxyl of mannose is clearly visible in the density at subsite +1. The pyranose O5-imidazole ring interaction (3.6 Å) is indicated with dashes. The interaction of MeHis and the mannosyl residue is very similar to the interaction with glucosyl residues in previous complexes[38]. A 2$F_{obs}$–$F_{calc}$ electron density map is shown at 1σ contour level for panels **a**, **d** and **f**

using EPR spectroscopy (Table 3) to investigate the electronic state of the active site copper upon binding. As has been shown by Frandsen et al.[33] and Courtade et al.[21], the binding affinity of oligosaccharide substrates is significantly affected by the presence of the exogenous ligand on the copper ion. Accordingly, EPR experiments were carried out in both the absence and presence of 200 mM chloride (1.0 M chloride for Xyl₆ studies). Furthermore, experiments were carried out at high substrate concentration to maximize substrate binding. For *Ls*AA9A a wide range of substrates was tested. In all cases, the parallel region of the spectra could be modelled with reliable $g_z$ and $|A_z|$ values, giving some insight into the electronic nature of the copper ion. Perpendicular values were less reliable due to the second-order nature of the spectra in this region, and are therefore not used in the analysis, although the appearance of superhyperfine coupling to ligands in this region was apparent in some cases (Table 3, Supplementary Fig. 14) and used as an indication of increased metal-ligand covalency in the singly-occupied molecular orbital (SOMO), as previously discussed by Frandsen et al.[33]. In all cases apart from xylan, the addition of substrate gave perturbation of the Cu spin Hamiltonian parameters similar to that already reported by Frandsen et al.[33]. In particular, shifts in $g_z$ values to ca 2.23 (along with the appearance of strong superhyperfine coupling) were seen upon addition of Avicel, glucomannan and xyloglucan, indicative of chloride coordination to the copper ion in the equatorial position of the copper coordination sphere. These shifts are analogous to those of *Ls*AA9A interacting with Cell₆ and PASC[33]. In contrast, addition of Xyl₆ did not give significant shifts in $g_z$ but did give perturbations in the $|A_z|$ value, with the appearance of superhyperfine coupling indicative of a second species different from that formed with Cell₆. The EPR spectra of *Ls*AA9A binding to Xyl₆ and xylan are indicative of substrate binding to the enzyme (although binding of Xyl₆ could be achieved only at high chloride concentrations), but without the chloride occupying the equatorial coordination position on the copper ion, revealing that these substrates drive an electronic state at the copper ion that is different to that of the other substrates. EPR perturbation was seen upon addition of Cell₆ to *Cv*AA9A but not with Xyl₆, consistent with the observed activity on Cell₆ and not Xyl₆ (Fig. 7; Supplementary Fig. 15).

## Discussion

Our understanding of the molecular basis for substrate binding and cleavage has been aided by the recent report of a crystal structure of *Ls*AA9A in complex with Cell₃ and Cell₆, as well as biochemical and EPR data for *Ls*AA9A on cellulosic substrates[33]. Here we have extended this biochemical, EPR and structural analysis by using a range of substrates as well as an additional related enzyme, *Cv*AA9A, to provide a better insight into substrate specificity.

Extensive probing of *Ls*AA9A and *Cv*AA9A substrate specificity showed that both cleave a range of cellulosic and non-cellulosic substrates, some of which have been shown for other AA9s[19, 20, 22, 23, 27, 29, 35, 39–41]. We made a number of important observations. Notably, whereas LPMO activity on cellulose-associated xylan has been previously observed for *Mt*LPMO9A[23], we report activity of *Ls*AA9A on xylo-oligosaccharides and isolated xylan; this may have important implications for the use of LPMOs in biotechnological contexts. We also observe that both *Ls*AA9A and *Cv*AA9A are able to cleave glycosidic bonds adjacent to mannosyl residues (Supplementary Fig. 6), which occur interspersed randomly with glucosyl residues in glucomannan, a biochemical observation supported by the *Ls*AA9A:glucomannan oligosaccharide structure, which unambiguously shows a mannosyl residue at subsite +1. We also noticed substrate-specific

xylose at subsite +1 does not stack with MeHis1, and in fact appears to have no interactions with the enzyme, while the same residues that bind the subsite +2 glucosyl residue in the *Ls*AA9A:Cell₅ structure, Asn28, His66 and Asn67, interact here with O1, O5 and O1 of the +2 xyloside residue, respectively (Fig. 6c; Supplementary Table 3). A structure of *Ls*AA9A:Xyl₅ Cu (II) determined from a low X-ray dose data collection showed the substrate bound similarly, and revealed a mix of water/Cl⁻ in the axial position and a fully occupied equatorial water on the active site copper (Fig. 6b). Thus, in contrast to binding of cello- or glucomannan oligosaccharides, the axial water was not displaced by binding of Xyl₅.

**EPR data suggest alternative *Ls*AA9A substrate binding modes.** We studied substrate binding on both *Ls*AA9A and *Cv*AA9A

**Table 2 Scaled crystallographic data statistics and refinement statistics**

| | LsAA9A: Cell$_5$ | LsAA9A: G4G4G3G | LsAA9A:Xyl$_3$ | LsAA9A:Xyl$_4$ | LsAA9A:Xyl$_5$ | LsAA9A: Xyl$_5$ Cu(II) | LsAA9A:GM | CvAA9A |
|---|---|---|---|---|---|---|---|---|
| Data collection | | | | | | | | |
| Synchrotron | ESRF | MAX-lab | ESRF | ESRF | ESRF | ESRF | ESRF | MAX-lab |
| Beamline | ID23-1 | I911-2 | ID23-1 | ID23-1 | ID23-2 | ID23-2 | ID30-B | I911-3 |
| Wavelength ($\lambda$) | 0.97625 | 1.03841 | 0.97625 | 0.97625 | 0.87260 | 0.87260 | 0.96862 | 1.00000 |
| Space group | $P4_132$ | $P4_132$ | $P4_132$ | $P4_132$ | $P4_132$ | $P4_132$ | $P4_132$ | $P1$ |
| Cell dimensions | | | | | | | | |
| $a, b, c$ (Å) | 126.30 | 124.91 | 125.18 | 125.01 | 125.35 | 125.30 | 125.29 | 47.00, 59.42, 115.45 |
| $\alpha, \beta, \gamma$ (°) | 90 | 90 | 90 | 90 | 90 | 90 | 90 | 102.67, 98.89, 89.54 |
| Resolution (Å) | 50.00-1.75 (1.80-1.75)[a] | 50.00-2.00 (2.05-2.00) | 50.00-1.50 (1.54-1.50) | 44.20-1.59 (1.70-1.59) | 50.00-1.33 (1.36-1.33) | 50.00-1.90 (1.95-1.90) | 50.00-1.48 (1.52-1.48) | 50.00-1.90 (1.95-1.90) |
| $R_{meas}$ | 0.118 (1.81) | 0.238 (1.64) | 0.114 (1.67) | 0.097 (1.84) | 0.098 (2.62) | 0.289 (1.58) | 0.133 (3.24) | 0.124 (0.828) |
| $I/\sigma I$ | 11.53 (1.02) | 12.31 (1.85) | 16.84 (1.81) | 17.53 (1.92) | 20.13 (1.37) | 6.30 (1.37) | 24.72 (1.50) | 6.70 (1.22) |
| Completeness (%) | 99.1 (98.4) | 99.9 (99.9) | 99.7 (96.9) | 99.8 (100) | 100 (100) | 99.8 (97.7) | 100 (100) | 96.1 (92.7) |
| CC(½) (%) | 99.7 (40.4) | 99.6 (49.8) | 99.8 (51.6) | 99.9 (61.8) | 100 (51.6) | 98.8 (62.5) | 100 (57.7) | 99.1 (64.3) |
| Observed reflections | 361,920 (25,195) | 304,791 (19,136) | 560,310 (53,847) | 475,945 (87,679) | 1,673,125 (123,274) | 234,494 (16,918) | 3,982,032 (199,606) | 196,670 (10,957) |
| Unique reflections | 34,937 (2502) | 23,096 (1680) | 36,460 (3782) | 45,264 (8102) | 77,258 (5639) | 27,034 (1914) | 105,966 (7879) | 90,992 (6493) |
| Redundancy | 10.36 (10.07) | 13.20 (11.39) | 10.41 (9.64) | 10.51 (10.82) | 21.66 (21.88) | 8.67 (8.84) | 37.59 (25.33) | 2.16 (1.69) |
| Refinement Resolution (Å) | | | | | | | | |
| No. mol. ASU | 1 | 1 | 1 | 1 | 1 | 1 | 1 | 6 |
| $R_{work}/R_{free}$ (%) | 18.15/ 21.45 | 17.30 /21.09 | 13.76/17.18 | 16.02/17.55 | 11.48/14.27 | 16.40/20.71 | 12.32/17.04 | 19.79 / 24.44 |
| No. of atoms | | | | | | | | |
| Protein[b] | 1821 | 1885 | 1884 | 1838 | 1882 | 1818 | 1869 | 10,673 |
| Ligand/ion | 61 | 60 | 100 | 66 | 56 | 56 | 176 | 40 |
| Water | 247 | 270 | 374 | 262 | 449 | 403 | 601 | 1041 |
| B-factors | | | | | | | | |
| Protein[b] | 32.1 | 23.9 | 17.8 | 25.0 | 18.0 | 18.4 | 22.8 | 26.2 |
| Ligand/ion | 47.8 | 35.9 | 36.0 | 52.9 | 24.0 | 27.8 | 34.2 | 70.6 |
| Water | 42.7 | 31.0 | 33.5 | 39.6 | 37.9 | 32.0 | 39.2 | 28.1 |
| R.m.s. deviations | | | | | | | | |
| Bond lengths (Å) | 0.0135 | 0.0183 | 0.0186 | 0.0174 | 0.0245 | 0.0167 | 0.026 | 0.0141 |
| Bond angles (°) | 1.5018 | 1.9101 | 1.8982 | 1.7407 | 2.0278 | 1.7393 | 2.3372 | 1.7245 |

[a]Highest-resolution shell is shown in parentheses
[b]Glycosylation (a single N-acetylglucosamine unit) and the active site copper are included in 'Protein'

oxidation profiles, namely that LsAA9A and CvAA9A cleaved small oligosaccharides using a C4-oxidising mechanism whereas they cleaved polysaccharides with both C1- and C4-oxidising mechanisms in varying proportions. Assuming a copper-based oxidative species, the similar distances between both C1 and C4 axial protons and the active oxygen species, as noted in Frandsen et al.[33], may allow slight differences in substrate binding to switch the C–H bond that is closest to attack. Substrate binding differences may also subtly alter the electronics at the copper site, which potentially could also favour a specific oxidation site. Oxidation regioselectivity is therefore less likely to be a strong functional constraint. This is in agreement with the presence of C1 and C4 regiospecificity in several clades in the N-terminal sequence similarity tree (Fig. 1). On the other hand, the tree enabled us successfully to predict that LsAA9A and CvAA9A might have similarities in having activity on a range of soluble substrates.

Our investigation highlighted many examples of the way in which substrate specificity and the site of attack on a polysaccharide is dictated by binding cleft interactions with the substrate. For example, LsAA9A's preference to cleave Cell$_4$ into Cell$_2$ (as shown for NcLPMO9C[18]), compared to the product profile of CvAA9A of Cell$_3$, Cell$_2$ and Glc (Supplementary Fig. 5), could be attributed to binding cleft interactions at the +2 subsite. This is because, though CvAA9A shares the same fold, active site co-ordination and overall structure with LsAA9A, it lacks all three of the subsite +2 substrate-binding residues in LsAA9A (Asn28, His66 and Asn67) (Supplementary Figs. 1 and 10). This suggests that, while LsAA9A binds Cell$_4$ from subsite −2 to +2, CvAA9A binds between subsites −3 and +1.

The structural data provide a molecular rationale for how LsAA9A is able to catalyse the unexpected cleavage of mannose-containing bonds. The LsAA9A:glucomannan oligosaccharide crystal structure shows the presence of a mannose (the C2 epimer of Glc), and essentially no Glc, at subsite +1. Talose (C4 epimer of mannose) and galactose (C4 epimer of Glc) arose from the reduction of C4-oxidised cleavage products (Supplementary Fig. 6), also suggesting both mannosyl and glucosyl occupation of

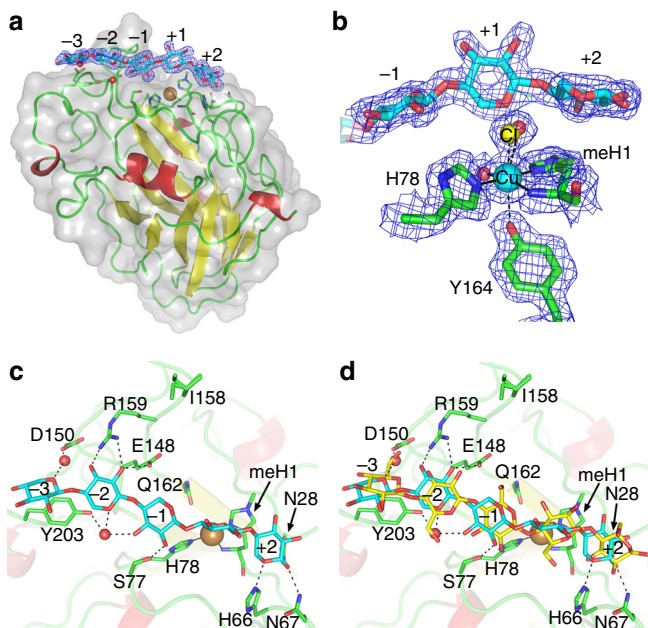

**Fig. 6** Structure of the *Ls*AA9A:xylo-oligosaccharide complex. **a** *Ls*AA9A: Xyl₅ with bound substrate (in cyan) from subsite −3 to +2. **b** Active site structure in the low-dose *Ls*AA9A:Xyl₅ Cu(II) structure, showing that the +1 xylosyl unit does not directly interact with the enzyme or displace the axial ligand on the copper (modelled as chloride and water in 0.5:0.5 ratio). **c** Top-down view *Ls*AA9A:Xyl₅ (in cyan). **d** Top-down view *Ls*AA9A:Xyl₅ (in cyan) and *Ls*AA9A:Cell₅ (in yellow) shown for comparison. A $2F_{obs}−F_{calc}$ electron density map is shown at 1σ contour level in panels **a** and **b**

subsite +1. We have previously described that the glucosyl unit at the +1 subsite in *Ls*AA9A Cell₃ and Cell₆ complexes[33,38] interacts with MeHis1 through its β-face, and while glucose can make carbohydrate-aromatic stacking interactions[42] through both faces of the pyranose ring, β-mannose is believed to have absolute preference for interactions through its α-face due to its axial C2-hydroxyl. Nonetheless, determination of the crystal structure of glucomannan fragments with *Ls*AA9A confirmed that this type of interaction takes place, and the mannosyl residue at the +1 subsites interacts with MeHis through its β-face with an O5-imidazole ring centre distance of 3.6 Å (Fig. 5f) (compared to 3.4–3.5 Å for the cello-oligosaccharide complexes). No similar interactions could be found through a search in the PDB. We did not observe a mannosyl residue at the −1 subsite in the structure, but it would cause no steric clash and so could be readily accommodated (though it would cause the loss of a hydrogen bond interaction with Ser77).

Although we were unable to obtain a structure with xyloglucan oligosaccharides bound, our observation that both *Ls*AA9A and *Cv*AA9A cleaved xyloglucan DP14–18 oligomers with the sole unsubstituted backbone glucosyl residues at subsite +1 (XXX/GXXXGol; as found for *Nc*LPMO9C[19]) is consistent with the binding of xyloglucan's cellulosyl backbone being similar to the binding of cello-oligosaccharides. This would suggest that the *Ls*AA9A could tolerate glucosyl residues with C6 xylosyl substitutions at subsites −1 or +2, but not at +1 where the C6 hydroxymethyl group occludes the copper axial binding site, and displaces the axial water.

*Ls*AA9A under the selected conditions degraded Xyl₆ with about 1/100 the efficiency as Cell₆, while *Cv*AA9A left Xyl₆ essentially untouched at all conditions tested (Supplementary Fig. 9). The differences in key amino acids involved in defining the *Ls*AA9A and *Cv*AA9A subsites, particularly the +2 subsite (vide supra) (Supplementary Fig. 1 and Supplementary Fig. 10),

are likely an important factor in *Ls*AA9A's superior xylan-degrading activity.

Our observation in *Ls*AA9A complexes that MLG oligosaccharides were unable to bind with β-(1 → 3)-glucan bonds near the active site are consistent with our observation that both *Ls*AA9A and *Cv*AA9A favour the cleavage of cellulosyl regions in MLG.

Not all aspects of substrate specificity could be explained through binding cleft interactions. Rather, aspects of the specificity differences appear to be mechanistic in origin and relate to the reactivity of different substrates. The high activity, spectroscopy data and structures of *Ls*AA9A and *Cv*AA9A with β-(1 → 4)-glucan substrates leads us to suggest the effective oxidative mechanism deployed in these situations may be regarded as a 'canonical pathway'. It is clear, however, that LPMOs may also have other 'non-canonical pathway' mechanisms, as exemplified by the differences between binding, spectroscopy and structures of *Ls*AA9A with Xyl₆. The crystallographic and EPR data show that a chloride ion—an oxygen species mimic—is not recruited into the copper's equatorial binding site upon xylooligosaccharide substrate binding, as happens in the canonical mechanism described by Frandsen et al.[33] The aldopentose nature of xylose categorically excludes the synergistic binding of saccharide ligand and molecular oxygen, which is brought about by a bridging 'pocket' water molecule between the C6-hydroxymethyl group of Glc and the amino terminus of the enzyme. This suggests a different oxidative mechanism may well be in operation for the cleavage of xylose-based substrates by *Ls*AA9A. Indeed, as has already been proposed by Kjaergaard et al.[43], activation of O₂ by an AA9 from *Thermoascus aurantiacus* probably gives formation of a copper-bound superoxide or hydrosuperoxide (HO₂) through associative displacement of a superoxide anion by a water molecule through the axial coordination site on the copper ion. In particular, a superoxide ion bound to the copper in the axial position would be in position to cleave a saccharidic chain by direct attack. Such a mechanism is expected when the axial water molecule on the copper ion is not displaced by the binding of substrate, as is the case with the binding of Xyl₅ to *Ls*AA9A. From the low-dose *Ls*AA9A:Xyl₅ structure described herein, the axial ligand is clearly present on the copper ion, though it is best modelled as a mixture of chloride and water, and the Tyr-O distance (2.86 Å) is not shortened compared to the un-complexed low dose structure (2.72 Å—PDB 5ACG). This is in contrast to the low dose *Ls*AA9A:Cell₃ structure where the Tyr-O distance is 2.47 Å (PDB 5ACF). Furthermore, the equatorial position in the low dose *Ls*AA9A:Xyl₅ is occupied by a water molecule, not a chloride ion, as corroborated by the EPR spectroscopy. Thus, a mechanism by which a copper-bound superoxide is generated next to the substrate is possible within the *Ls*AA9A-Xyl₅ complex. Such a mechanism may be expected to be rate-independent on the redox potential of the reducing agent, since the rate-limiting step is likely to be hydrogen atom abstraction by the superoxide from the substrate rather than reductive cleavage of the O–O bond. Therefore, the fact that the rate of cleavage of xylan and Xyl₆ by *Ls*AA9A is less dependent on reducing agent while the cleavage of the other substrates is strongly dependent (Fig. 4) illustrates that a different oxidative mechanism is in operation. Thus the extent of activity on certain substrates is a function of the oxidative species which can be formed at the copper ion which is—in turn—dependent on the substrate. This means that for some substrates the use of reducing agents with different potentials can profoundly affect apparent substrate specificity. But, more importantly, LPMOs appear to have more than one oxidative mechanism available for substrate cleavage, governed to some extent by the nature of the substrate–LPMO interaction. Indeed, the existence of multiple oxidative mechanisms for a

**Table 3 Spin-Hamiltonian parameters (parallel region) for *Ls*AA9A and *Cv*AA9A in contact with substrates**

| Enzyme–substrate combination | $g_z$ | $A_z$ (MHz) | Comments |
|---|---|---|---|
| No NaCl | | | |
| *Ls*AA9A-$H_2O$ | 2.279 | 458 | Weak superhyperfine (SHF) coupling |
| *Ls*AA9A + Cell$_6$ | 2.273 | 515 | Intense SHF coupling |
| *Ls*AA9A + avicel | 2.278 | 470 | Weak SHF coupling |
| *Ls*AA9A + xylan | 2.272 | 480 | Spectrum complicated by organic-based radicals in perpendicular region |
| *Ls*AA9A + glucomannan | 2.232 | 518 | Very likely NaCl contamination in the substrate. Intense SHF coupling |
| *Ls*AA9A + xyloglucan | 2.270 | 515 | Very intense SHF coupling. |
| 200 mM NaCl | | | |
| *Ls*AA9A-Cl | 2.258 | 455 | Likely mixture of $H_2O$ and Cl species. |
| *Ls*AA9A + Cell$_6$ | 2.234 | 517 | Intense SHF coupling |
| *Ls*AA9A + avicel | 2.232 | 522 | Slight change in perpendicular region, some appearance of SHF coupling |
| *Ls*AA9A + xylan | 2.270 | 470 | Spectrum complicated by organic-based radicals in perpendicular region |
| *Ls*AA9A + solubilised xylan | 2.272 | 470 | Radical impurities present in the perpendicular region |
| *Ls*AA9A + glucomannan | 2.231 | 520 | Intense SHF coupling |
| *Ls*AA9A + solubilised glucomannan | 2.233 | 515 | Intense SHF coupling |
| *Ls*AA9A + xyloglucan | 2.228 | 530 | Intense SHF coupling |
| *Ls*AA9A + Xyl$_6$ | 2.268 | 400 | Very rhombic, different from both Cell$_6$-bound and unbound protein, intense SHF coupling. Could only be achieved with very high Xyl$_6$ concentrations. |
| *Cv*AA9A | 2.273 | 476 | Likely no Cl species present |
| *Cv*AA9A + Cell$_6$ | 2.228 | 527 | Mixture of C$_6$-bound and unbound *Cv*AA9A. Some SHF coupling visible. Full binding could not be achieved even with large excess of Cell$_6$ |
| *Cv*AA9A-1 M NaCl | 2.273 | 468 | Likely no Cl species present even in the presence of 1 M NaCl |
| *Cv*AA9A + Xyl$_{6-1}$ M NaCl | 2.273 | 468 | Spectrum identical to the unbound form, even at very high Xyl$_6$ concentrations. |

The experiments were performed with or without 0.2 M chloride. For xylohexaose, 1.0 M chloride was used. Spectra are shown in Supplementary Figs. 14 and 15. Due to the high amount of protein required by the technique, the data presented are from single EPR experiments, although the spectra with Cell$_6$, Xyl$_6$ and avicel were performed in at least duplicate

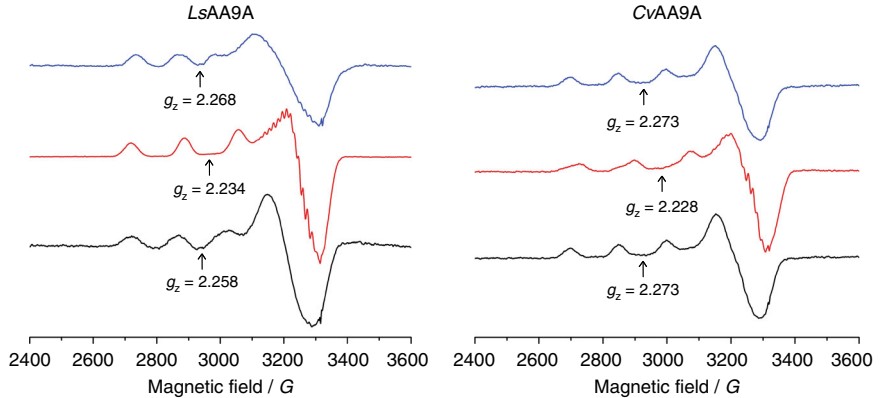

**Fig. 7** X band cw EPR spectra of *Ls*AA9A (left) and *Cv*AA9A (right), 150 K. Spectra were collected in the presence of 1 M NaCl (black), Cell$_6$ and 200 mM NaCl (red), or Xyl$_6$ and 1 M NaCl (blue). Spectra are representative of duplicate experiments

single LPMO is an intriguing contribution to the ongoing debate about LPMO mode of action.

Results obtained in this study broaden the known substrate specificity of AA9 LPMOs to include isolated xylan and xylo-oligosaccharides, and mannosyl-containing bonds within glucomannan. We further show that oxidation type (C1/C4) is influenced by substrate type, and in this work differed between oligo- and polysaccharides. This investigation into the molecular causes of AA9 LPMO substrate specificity demonstrated the existence of multiple influences. As with carbohydrate-acting hydrolases, for example, LPMO substrate specificity is dictated by binding cleft protein:carbohydrate interactions. But in addition, the fact that activity on some substrates is differentially responsive to reducing agent potential suggests that these carbohydrates do not properly activate the active site copper, and are cleaved through an alternative oxidative pathway. Combinations of canonical and non-canonical mechanisms greatly extend the range of potential substrates for LPMOs and offer new insight into their biochemical mode of action.

## Methods

**Phylogenetic tree**. AA9 is a family with more than 6000 sequences listed in NCBI nr and JGI databases in 2016. Because of high variability in the N-terminal portion of LPMO amino-acid sequences, no significant global alignment of LPMOs can be obtained—thereby limiting global downstream phylogenetic analyses. We chose therefore to extract the highly variable N-terminal half of these sequences (which includes two histidine residues involved in the coordination of the copper atom) for phylogenetic analysis as well as to limit the analysis to sequences that are closely related to each other and to those that have been biochemically characterized in the literature. We reduced the set of AA9 sequences to those that gave BLAST bit-scores greater than or equal to a value of 200, using *Ls*AA9A and *Ta*AA9A as queries. A Jaccard distance matrix was compiled from BLAST bit scores and represented as a tree, built according to the principle of neighbor-joining method[44] displaying the resulting 444 sequences (Fig. 1).

**Protein production**. The gene encoding *Ls*AA9A was amplified from genomic DNA of *Lentinus similis* and the gene encoding *Cv*AA9A was amplified from genomic DNA of *Collariella virescens* (formerly known as *Chaetomium virescens*). Both were cloned in *E. coli* using primers containing insertion sites for the vector pDau109 used for cloning. The fragments were then cloned into BamHI- and XhoI-digested pDau109 using an IN-FUSION Cloning Kit. Cloning of the genes into BamHI- and XhoI-digested pDau109 resulted in transcription of the recombinant

enzymes encoding gene under the control of a NA2-tpi double promoter. The treated plasmids and inserts were transformed into One Shot TOP10F Chemically Competent E. coli cells (Invitrogen) according to the manufacturer's protocol, spread onto LB plates supplemented with 0.1 mg/ml ampicillin and incubated at 37 °C overnight. Colonies of each transformation were cultivated in LB medium supplemented with 0.1 mg/ml ampicillin and plasmids were isolated using a QIAPREP Spin Miniprep Kit (Qiagen).

LsAA9A and CvAA9A was expressed in Aspergillus oryzae MT3568. Transformants producing the recombinant enzymes were inoculated in 2 l of Dap-4C medium and incubated at 30 °C for 4 days. Mycelia were removed by filtration, and the medium was collected for purification. Ammonium sulfate was added to the sterile filtered medium to a conductivity of 200 mSi/cm and the pH adjusted to 7.5. The broth was applied to a 50/15 Butyl Toyopearl column (Tosoh Biosciences) equilibrated with 25 mM Tris, 1.5 M ammonium sulfate, pH 7.5. The column was washed in the same buffer and eluted with a gradient to 25 mM Tris, pH 7.5. Fractions containing recombinant enzymes were combined and washed with milliQ water by ultrafiltration (10 kDa MWCO, PES filter, Sartorius) to a conductivity of 1.2 mSi/cm. The pH was adjusted to 8.0 and applied to a 50/40 Q Sepharose FF column (GE Healthcare) equilibrated with 20 mM Tris, pH 8.0. The column was washed in the same buffer and the enzyme eluted with a gradient from 0 to 0.5 M sodium chloride. Fractions containing LsAA9A were combined and concentrated by ultrafiltration using VIVASPIN 20 (10 kDa MWCO) spin concentrators.

**Enzyme assays**. Apo-LsAA9A and apo-CvAA9A were pre-incubated for 0.5–1 h at 5 °C in 0.9 stoichiometric Cu(II)(NO$_3$)$_2$ immediately before enzyme reactions. AA9 enzyme reactions on oligosaccharides were in 10 µl containing 5 nmol oligo-saccharide, 100 mM ammonium formate pH 6, ±4 mM ascorbate, pyrogallol or cysteine, ±5 pmol LsAA9A or TaAA9A and were incubated at 20 °C for 4 h. Xyloglucan endoglucanase (XEG) reactions were in 10 µl containing 5 nmol oligosaccharide, 100 mM ammonium formate pH 6, ±10 µmol GH5 XEG and were incubated at 20 °C for 4 h. Oligosaccharides were purchased from Megazyme (see also following section). In general, enzyme reactions on polysaccharides were in 100 µl containing 0.5% (w/v) polysaccharide, 100 mM ammonium formate pH 6, ±4 mM ascorbate, pyrogallol or cysteine, ±63 pmol LPMO, and were incubated at 20 °C for 16 h. Avicel cellulose was purchased from Sigma-Aldrich, UK; barley beta-glucan medium viscosity (mixed-linkage glucan), konjac glucomannan, tamarind xyloglucan, birchwood xylan, corn starch and laminarin were purchased from Megazyme, Ireland; squid-pen β-chitin was a kind gift from Dominique Gillet of Mahtani Chitosan. Phosphoric acid-swollen cellulose (PASC) was prepared by making a slurry of 1 g Avicel cellulose (Sigma-Aldrich) with 3 ml H$_2$O before adding 30 ml ice-cold phosphoric acid and incubating at 0 °C for 1 h. The cellulose was then washed numerous times with water until the flowthrough had a neutral pH before use in reactions. Mixed-linkage glucan, glucomannan, xyloglucan, xylan, starch and laminarin were boiled for 5 min to make solubilized 1% (w/v) stock solutions before reactions. To aid solubilisation where necessary, water was added to a methanol: polysaccharide slurry before boiling, which improved dispersion throughout the water. Reactions were routinely stopped by addition of three reaction volumes of 96% (v/v) ethanol before precipitation of the undigested substrates, and separation of the reaction products for further analysis. For poly-saccharide analysis by carbohydrate electrophoresis (PACE), reaction products and oligosaccharide standards (Megazyme) were reductively aminated with 8-amino-naphthalene-1,3,6-trisulfonic acid (ANTS; Invitrogen, http://www.lifetechnologies.com) and separated by acrylamide gel electrophoresis. In all cases, an Hoefer SE 660 vertical slab gel electrophoresis apparatus (Amersham, Buckinghamshire, UK) was used with 24-cm plates, 0.75-mm spacer, and well of width 0.25 cm. Standard glass or low-fluorescence Pyrex plates were used. Electrophoresis was performed at 10 °C in all cases. High concentration gel PACE was performed using a 192 mM glycine, 25 mM Tris, pH 8.5, running buffer. The gel contents were as follows: resolving gels, 37.5 ml 40% (w/v) acrylamide, 12.5 ml 375 mM Tris-HCl buffer, pH 8.8, 100 µl 10% ammonium persulfate, 50 µl tetramethylethylenediamine (TEMED); stacking gels, 2 ml 40% (w/v) acrylamide, 2.5 ml 375 mM Tris-HCl buffer, pH 8.8, 5.4 ml water, 100 µl 10% ammonium persulfate, 10 µl TEMED. Electrophoresis was carried out at 100 V for 30 min, 500 V for 30 min and 1000 V for 180–210 min, and gels were then visualized with a G-box (Syngene) equipped with a short pass detection filter (500–600 nm) and long-wave UV tubes (365 nm emission). Low concentration gel PACE was performed using a 0.1 M Tris–borate pH 8.2 buffer system. The gel contents were as follows: 20% (w/v) polyacrylamide gel contained 0.5% (w/v) N,N 9-methylenebisacrylamide with a stacking gel (2 cm) of 8% (w/v) polyacrylamide and 0.2% (w/v) N,N 9-methylenebisacrylamide. The samples were electrophoresed initially at 200 V for 20 min and then at 1000 V for 90 min. All experiments were carried out at least in triplicate. See Supplementary Fig. 16 for uncropped gels scans.

Sodium borohydride reducing agent experiments were performed by addition of ammonia to a concentration of 2 M to reaction mixtures. After incubation for 5 mins, 1/20 solution volumes of 10% (w/v) NaBH$_4$ 2 M ammonia were added, before incubation for 16 h at 25 °C. Samples were dried in vacuo and redissolved in, and redried from, 100 µl 10% (v/v) acetic acid 90% (v/v) methanol, five times. HPAEC was performed on a CarboPac PA1 column (Dionex) with injections of 20 µl and elution at 0.4 ml min$^{-1}$. The elution profile was: 0–3 min, 10 mM NaOH

(isocratic); 3–6 min, 10 → 1 mM NaOH (linear gradient); 6–19 min, 1 mM NaOH (isocratic); 19–37 min, 45 mM NaOH, 225 mM sodium acetate (isocratic). A pulsed amperometric detector (PAD) with a gold electrode was used. PAD response was calibrated using markers (500 pmol).

**X-ray crystallography and PDB database searches**. All crystallization trials were set up in MRC two-well plates at room temperature using an Oryx-8 robot (Douglas Instrument). Crystals were obtained by sitting-drop vapour diffusion technique in drops of 0.3–0.5 µl with a reservoir volume of 100 µl. Pre-incubation with 1–2 mM Cu(II) acetate for 30–60 min was carried out for all crystallization trials. Crystallization and post-crystallization experimental details are shown in Supplementary Table 4. Crystals were cryocooled in liquid nitrogen and all datasets were collected at cryogenic temperatures (100 K) at either the MX beamlines I911-2/I911-3 at MAX-lab in Lund, Sweden, or at the MX beamlines ID23-1, ID23-2 or ID30-B at ESRF, Grenoble, France (Table 2). All crystals were obtained by vapour diffusion technique set up in sitting-drop MRC plates, with a reservoir volume of 100 µl and at room temperature using an Oryx-8 robot (Douglas Instrument). All data were collected at cryogenic temperatures (100 K) after cryocooling the crystals in liquid nitrogen. Oligosaccharide substrates used for soaking were purchased from Megazyme (MLG (G4G4G3G and G4G3G4G), xylotriose (Xyl$_3$), xylotetraose (Xyl$_4$), xylopentaose (Xyl$_5$), xyloglucan heptasaccharide (XXXG), cellotriose (Cell$_3$) or provided by Novozymes A/S (cellopentaose (Cell$_5$)). Data were initially collected on crystals soaked with G4G4G3G (LsAA9A:G4G4G3G; PDB 5NLR), Xyl$_3$ (LsAA9A:Xyl$_3$; PDB 5NLQ), Xyl$_4$ (LsAA9A:Xyl$_4$; PDB 5NLP) and Cell$_5$ (LsAA9A: Cell$_5$; PDB 5NLS). On a crystal soaked with Xyl$_5$ a data set with reduced X-ray dose ((LsAA9A:Xyl$_5$Cu(II); PDB 5NLN; 40 frames of 5.7% transmission, 0.05 s expo-sure/frame, 1° oscillation with a beamsize of 10 × 10 µm) was collected using helical collection to minimize photoreduction of the active site copper. Subsequently, on similar crystals another full dose data set was collected to high resolution (LsAA9A: Xyl$_5$; PDB 5NLO). Ladders of glucomannan (GM), from konjac, and of xyloglucan (XG), from tamarind, were prepared by partial acid hydrolysis (20–200 mM TFA for 20 min at 120 °C) of polysaccharide substrates purchased from Megazyme. Hydrolyzed products were isolated using ethanol precipitation to remove the remaining polysaccharides. The oligosaccharides were dried thoroughly using a SpeedVac. Data were collected on crystals soaked in GM (LsAA9A:GM; PDB 5NKW) or XG stock solutions (in 3.8 M NaCl, 0.1 M citric acid pH 5.5). Crystals were also soaked in the presence of 0.3 M XXXG, and up to 1.2 M of XG oligo-saccharide purchased from Megazyme (consisting primarily of XXXGXXXG, see Courtade et al.[21]). No complex structures were obtained from any of the crystals soaked with XG substrates, either because no binding was observed or only cel-looligosaccharides were bound (presumably because acid hydrolysis caused deb-ranching). A stereo figure for typical density in the lowest resolution LsAA9A: oligosaccharide complex is shown in Supplementary Fig. 12.

CvAA9A was deglycosylated in 20 mM MES, pH 6.0, 125 mM NaCl by incubation with ~0.03 units per mg CvAA9A of endoglycosidase H from (Roche Diagnostics, 11643053001), and then buffer exchanged to 20 mM Na-acetate pH 5.5. Intergrown crystals were initially obtained in an index screen in conditions of 1.5–2.0 M (NH$_4$)$_2$SO$_4$ (and in some cases 0.1 M NaCl) in pH 6.5–8.5 (0.1 M of either Bis-Tris, HEPES or Tris). The crystals diffracted to 2.0–3.5 Å resolution but were multiple. Crystal conditions were optimized in a range of 1.2 M–2.6 M (NH$_4$)$_2$SO$_4$ (±0.1 M NaCl) in pH 6.5–8.5, which produced crystal plates suitable for mounting. A data set collected at I911–3 on a crystal grown in 0.1 M Bis-Tris pH 6.5, 2.0 M (NH$_4$)$_2$SO$_4$ could be processed in $P2_1$ to 2.5 Å. A preliminary CvAA9A structure with four molecules in the asymmetric unit was solved by Molecular Replacement using MOLREP with modified coordinates of the high-resolution structure of LsAA9A (PDB 5ACH), which is 41% identical, as a model and refined isotropically to an $R_{free}$ of 32%. From another data set (collected on a crystal grown in presence of 0.1 M NaCl; Supplementary Table 4) a structure solved (using the preliminary one) with six molecules in the asymmetric unit could be fully modelled and refined resulting in the complete CvAA9A structure (PDB 5NLT; Table 2). The six molecules in the asymmetric unit are very similar (average RMSDs of 0.08 Å). The density of MeHis1 is less clear in chains C and F, and in particular methylation is not as obvious in all chains. Soaks (with 1.2 M Cell$_3$) of CvAA9A were also prepared. Data were collected to 2.1 Å and the electron density showed a Cell$_3$ molecule, which however was not bound at the active site. Soaks with Cell$_6$ damaged the crystals.

Each data set was processed using XDS[45] (the resolution cutoff was chosen on the basis of a CC½ around 50%) and subsequently scaled using XSCALE. Refmac[46] was used for restrained refinement of the structures in which LsAA9A: Xyl$_5$ was refined anisotropically, while LsAA9A:Xyl$_3$ was refined anisotropically for protein atoms and isotropically for all other atoms. All other structures were refined isotropically for all atoms. For LsAA9A:G4G4G3G the structure was best modelled by the G4G4G portion of the substrate bound mainly in subsite –1 to +2 (80% occupancy) and with a minor conformation occupying subsite –2 to +1 (20% occupancy). Near subsite, –2 a number of water molecules were modelled with 80% occupancy. Ligands and structures were modelled in COOT[47] and validated using MolProbity (within COOT) and Procheck (CCP4 suite), which reported Ramachandran plots with 99% of residues in allowed regions for all structures. Scaled data statistics and refinement statistics are summarized in Table 2.

To identify potential stacking interactions of the β-face of β-mannose with His, the PDB database was searched with Glyvicinity[48]. First all protein/β-mannose

interactions within a distance cut-off of 4.0 Å for structures determined at a resolution better than 3.0 Å were identified. Among these, only two structures were found where the interactions involved His residues and the pyranose O5. The interactions between the imidazole and the pyranose rings were side by side or almost perpendicular, and thus not comparable with the +1 subsite interactions of the *Ls*AA9A complexes.

**EPR spectroscopy.** Continuous wave (cw) X-band frozen solution EPR spectra of 0.2–0.3 mM solution of *Ls*AA9A or *Cv*AA9A (in 10% v/v glycerol) at pH 6.0 (50 mM sodium phosphate buffer with or without addition of 200 mM NaCl or 20 mM MES buffer, 200 mM NaCl) and 165 K were acquired on a Bruker EMX spectrometer operating at ~9.30 GHz, with modulation amplitude of 4 G, modulation frequency 100 kHz and microwave power of 10.02 mW (three scans). Avicel cellulose, konjac glucomannan, tamarind xyloglucan and birchwood xylan were added to the EPR tube containing the protein as solids. Alternatively, glucomannan and xylan were heated until dissolution (ca. 2 min) to make solubilized 1% (w/v) stock solutions in water, which were then used for addition of excess polysaccharide to *Ls*AA9A. Cellohexaose and xylohexaose were added to the protein solution either from stock solutions in water or as a solid up to 60- or 150-fold excess, respectively. For the experiments in the presence of xylohexaose, additional NaCl was added to the protein alone or the protein:Xyl$_6$ mixture from a 5 M stock solution. Due to the high amount of protein required by the technique, the data presented are from single EPR experiments, although the spectra with Cell$_6$, Xyl$_6$ and avicel were performed in at least duplicate.

Spectral simulations were carried out using EasySpin 5.0.3[49] integrated into MATLAB R2016a[50] software on a desktop PC. Simulation parameters are given in Table 3. $g_z$ and $|A_z|$ values were determined accurately from the absorptions at low field. It was assumed that $g$ and $A$ tensors were axially coincident.

**Data availability.** Protein Data Bank: Atomic coordinates and structure factors for the reported crystal structures were deposited under accession codes 5NLT (*Cv*AA9A), 5NLS (*Ls*AA9A-Cell$_5$), 5NLR (*Ls*AA9A-G4G4G3G), 5NKW (*Ls*AA9A-GM), 5NLQ (*Ls*AA9A-Xyl$_3$), 5NLP (*Ls*AA9A-Xyl$_4$), 5NLO (*Ls*AA9A-Xyl$_5$) and 5NLN (*Ls*AA9A-Xyl$_5$-Cu$_{II}$), GenBank: Sequence data for *Cv*AA9A were deposited under accession code KY884985. Raw EPR data are available on request through the Research Data York (DOI: 10.15124/5810c962-148c-4328-ab92-895e2dae4d3c). The data that support the findings of this study are available from the corresponding author upon request.

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

## Acknowledgements

We thank MAX-lab, Sweden and the European Synchrotron Radiation Facility (ESRF), France, for synchrotron beamtime and assistance. Travel to synchrotrons was supported by the Danish Ministry of Higher Education and Science through the Instrument Center DANSCATT and the European Community's Seventh Framework Programme (FP7/2007-2013) under BioStruct-X (grant agreement 283570). This work was supported by the UK Biotechnology and Biological Sciences Research Council (grant numbers BB/L000423/1 to P.D. and P.H.W., and BB/L021633/1 to P.H.W.) and the Danish Council for Strategic Research (grant numbers 12-134923 to L.L.L. and 12-134922 to K.S.J.).

## Author contributions

T.J.S. carried out most of the activity assays, assisted by T.Tr. and L.F.L.W. K.E.H.F. carried out most of the structural studies with J.C.P., T.Ta. and L.L.L. L.C. carried out EPR spectroscopy. L.N. and B.H. carried out phylogenetic studies. M.T. and K.S. carried out target protein identification and production. P.D., L.L.L. and P.H.W. supervised the experimental work. T.J.S., K.E.H.F., K.S.J., P.H.W., L.L.L. and P.D. analysed the data and wrote the paper.

## Additional information

**Competing interests:** M.T. and K.S. are employees of Novozymes, a producer of enzymes for industrial use. The remaining authors declare no conflict of interest.

