## [Peer Review File · Nature Communications]

Reviewers' comments:

Reviewer #1 (Remarks to the Author):

The manuscript provides significant advances in understanding the substrate specificity of LPMOs, particularly as related to xylan (and derivatives) as a substrate, using a combination of biochemical, spectroscopic and crystallographic techniques to probe the influences of enzyme structure and active site metal environment. In my opinion, the data presented on the two phylogenetically and enzymatically distinct LPMOs are extensive and convincing in providing evidence for the conclusions drawn. This manuscript should be of broad interest to anyone who works with LPMOs, whether at the mechanistic level or using them to improve cellulase cocktails. A couple of comments/concerns:

1. I was not clear why the N-terminals of the enzymes were chosen for the phylogenetic analysis until I read 'Materials and Methods'. It would be useful to include some kind of explanation in 'Results' as well.
 2. The results of the EPR studies indicate that high NaCl concentration is required to observe binding of Xyl6 to LsAA9A (also, high NaCl concentrations are used in enzyme crystal preparation). Yet the assays are done in the absence of NaCl. Has it been determined whether NaCl affects the activities of the enzymes under study, in particular for xylan and xylooligosaccharides?
- And some minor suggestions:
- Figure 4a): are the ladders on both gels indeed the same (cello oligos)?
- Line 30: 'suggesting' for 'revealing'
- Line 95: 'insight into' for 'insight of'
- Line 183: 'neither enzyme can' for 'both enzymes cannot'
- Lines 440-441: 'Results obtained in this study.....' for 'in this study we broaden.....'
- Lines 446-449: In this sentence it seems to me that the explanation and the observation have been switched—would read better the other way around
- Line 472: 'ultrafiltration with 10 kDa cutoff' to 'ultrafiltration with 10 kDa cutoff filter'.

Reviewer #2 (Remarks to the Author):

This beautiful manuscript describes a detailed analysis of substrate specificity of CvAA9A, an LPMO enzyme identified through a bioinformatics search, and LsAA9A, a LPMO previously studied by this group (ref. 32). LPMOs are important accessory enzymes in the saccharification of lignocellulose: a key step in effective use of plant biomass for renewable energy. LPMOs catalyze the cleavage of polysaccharides via oxidative chemistry in a Cu-dependent mechanism using O₂ and reducing equivalents. Using a broad range of techniques including extensive product profiling, crystallography, and EPR spectroscopy, the authors show that CvAA9A has measurable activity against soluble xylan, a first in LPMOs and an activity of possible biotechnological interest. CvAA9A was selected for study as it has a different pattern of residues known to be important for h-bonding to the +2 glucosyl residue when compared to CvAA9A, which the authors thought may give a different substrate/product profile. In this manuscript, CvAA9A is also more extensively characterized in terms of substrate/product profile. The two enzymes are very similar in activity except with regard Xyl6, which LsAA9A oxidatively cleaves at ~ 1/100 the rate of Cell6. The EPR spectrum of LsAA9A bound to Xyl6 is significantly different to that of Cell6; in contrast CvAA9A suggests Xyl6 does not bind. This indicates a different electronic environment, and this is borne out by the crystal structures of the complexes of LsAA9A-Xyl5 and LsAA9A-Cell5, where Xyl does not directly ligate the copper, and chloride (as superoxide surrogate) binds axially rather than equatorially. The authors see an uncoupling of the rate of cleavage to reductant in the LsAA9A-xylan reaction, and suggest this is due to the superoxide being displaced by water and then attacking the xylan in a "free" state. Although I agree that initial Cu-reduction is no longer rate-limiting, I disagree that this equates to a model where superoxide must be displaced from the copper to react with the Xyl6. I think it much more likely that it remains activated by the copper,

and the rate is limited by attaining a productive reaction conformation/configuration with the non-biological substrate. Is there any mass spectrometry evidence that LsAA9A is more susceptible to oxidative damage with Xyl6 as substrate? This might indirectly support the authors' contention that superoxide must leave the copper to react with Xyl6, although the longer time frame required to productively cleave Xyl6 may lead to ROS damage to the enzyme anyway, as Cu(II)-superoxide will not be stable for long periods of time. However, this is a minor point of interpretation within an excellent manuscript in all other ways.

Overall, this manuscript adds significantly to the LPMO literature, and will be of broad interest not only to those interested in plant biomass degradation and LPMOs specifically, but to biochemists and chemists interested in copper enzymes, dioxygen activation, substrate specificity, and polysaccharide chemistry.

All my other comments are typographical or minor in nature.

(1) Line 213. I would refer to SI Tables S4-5 as well as Fig. 5 at this point.

(2) SI Fig. 1. Tyr169/164 is indicated as part of the His-brace in the legend, i.e. highlighted in yellow in the figure. Although a (sometimes) Cu ligand, it is generally not included in the definition of the His-brace. Either change the legend to state that yellow indicates Cu ligands rather than His-brace, or remove the yellow coloring from Tyr169/164.

(3) Line 401. I do not like the terms "on-pathway" and "off-pathway"; for xylan the pathway is not "off-pathway" for the mechanism of xylan cleavage, it is a different pathway. I would prefer "canonical pathway" and "non-canonical pathway" for polysaccharide cleavage.

(4) Line 498. "2-aminobenzamide" should be "2-Aminobenzamide" as it starts the sentence.

(5) Line 531 and 557. "Data was" should be "Data were".

(6) Line 548-550. It would be useful here to indicate the sequence identity between CvAA9A and LaAA9A.

(7) Line 561. In SI Table S5, the CC_1/2 is not above 50% for LsAA9A (40%). These data should be cut back.

(8) Line 599-600. There is no SI Table S2. I think the authors mean Table 2.

(9) Line 702. The final page is missing from ref 32; it should be 298-303.

(10) Fig 1a and Fig. S12o. I found it difficult to distinguish orange and red in these figure panels.

(11) Fig 5a, 7a,b, Fig. S11b. What type of electron density map is shown, and what is the contour level? Please add this information to legends.

(12) Fig. 6a, 7a. The electron density map mesh is too dense to make out the model. Either decrease the width of the mesh lines, or reduce the mesh density.

Reviewer #3 (Remarks to the Author):

In the paper by Simmons et al., entitled: "Structural and electronic determinants of lytic polysaccharide monooxygenase reactivity on polysaccharide substrates" the authors describe biochemical, biophysical and structural characterizations of two closely related Lytic polysaccharide

monooxygenases, LsAA9A from *Lentinus similis* and CvAA9A *Collariella virescens*.

The biochemical activity studies of the two LPMOs show that these cleave a range of different polysaccharides, including cellulose, xyloglucan, mixed-linkage glucan, and glucomannan. One of the two enzymes, LsAA9A, additionally cleaves isolated xylan substrates, this is the first LPMO characterized that show such activity.

The study also includes structural determination of the two studied LPMOs, including an impressive amount of protein ligand complex structures of LsAA9A. These ligand complexes of LsAA9A give detailed insights into the determinants of specificity by this LPMO. The ligand complex structures in combination with electron paramagnetic resonance characterizations of the two enzymes further reveal differences in copper co-ordination upon the binding of xylan compared to glucans. The results presented show that AA9 LPMOs can display different apparent substrate specificities dependent upon substrate type and the electronic state at the copper active site. It is worth noting that this is the second study ever that presents structures of a LPMO with a substrate molecule bound at the catalytic center of the enzyme. The previous study presenting LPMO ligand complex structures was presented partly by the same authors as in the current study.

Overall the study is very novel and includes an impressive amount of work and new results. The different types of studies included in the study are all well planned and well performed. The results from these studies are all of high standard and well-motivated to be included in the paper.

Major comments

The only major criticism of the results presented in the study is the part presenting the results from the phylogenetic analyses of 444 LPMO sequences. This part is neither very informative nor very novel. There are several previous papers presenting much more detailed phylogenetic analyses of LPMOs, including a study by some of the authors of this study. The authors should therefore strongly consider to either expanding the phylogenetic analyses part so that the results presented gives substantial new information compared with previous studies, or remove this part from the paper and instead include the results in the supplementary material section.

Minor comments

The author does use the definition binding sub sites at several places in the paper, e.g- subsite -4 to + 2, row 98, page 5, without giving the reader a good definition for this definition. For most readers, familiar with ligand complex structures of glycoside hydrolases this is a well-known way of defining how a ligand is bound in the substrate binding cleft/tunnel/pocket, but outside the GH research area this nomenclature is not as well-known and need a short clarification early on in the paper.

There are several abbreviations used in the text that should be better described before they are used. Examples of such are: Cell6-2AB, HPAEC, TFA hydrolysates, SOMO, PACE

Response to reviewers' comments on Simmons et al.

We thank the reviewers for their comments and the helpful suggestions.

Reviewer #1 (Remarks to the Author):

The manuscript provides significant advances in understanding the substrate specificity of LPMOs, particularly as related to xylan (and derivatives) as a substrate, using a combination of biochemical, spectroscopic and crystallographic techniques to probe the influences of enzyme structure and active site metal environment. In my opinion, the data presented on the two phylogenetically and enzymatically distinct LPMOs are extensive and convincing in providing evidence for the conclusions drawn. This manuscript should be of broad interest to anyone who works with LPMOs, whether at the mechanistic level or using them to improve cellulase cocktails.

A couple of comments/concerns:

1. I was not clear why the N-terminals of the enzymes were chosen for the phylogenetic analysis until I read 'Materials and Methods'. It would be useful to include some kind of explanation in 'Results' as well.

➤ We have provided a comment and reference in the results section to explain.

2. The results of the EPR studies indicate that high NaCl concentration is required to observe binding of Xyl6 to LsAA9A (also, high NaCl concentrations are used in enzyme crystal preparation). Yet the assays are done in the absence of NaCl. Has it been determined whether NaCl affects the activities of the enzymes under study, in particular for xylan and xylooligosaccharides?

➤ The requirement for chloride to detect binding of xylooligosaccharides in the EPR is very likely related to the effect on enhancing the binding of the substrate. This effect was explained in Frandsen et al. (2016). Chloride acts as a mimic for superoxide and the binding of chloride/superoxide with oligosaccharide substrate is synergistic. In the activity experiments the chloride is necessarily 'replaced' by O₂ and so the chloride can't be used to enhance productive binding, and hence chloride would not be expected to enhance activity of the LPMO.

And some minor suggestions:

➤ We made all the textual corrections.

Reviewer #2 (Remarks to the Author):

This beautiful manuscript describes a detailed analysis of substrate specificity of CvAA9A, an LPMO enzyme identified through a bioinformatics search, and LsAA9A, a LPMO previously studied by this group (ref. 32). LPMOs are important accessory enzymes in the saccharification of lignocellulose: a

key step in effective use of plant biomass for renewable energy. LPMOs catalyze the cleavage of polysaccharides via oxidative chemistry in a Cu-dependent mechanism using O₂ and reducing equivalents. Using a broad range of techniques including extensive product profiling, crystallography, and EPR spectroscopy, the authors show that CvAA9A has measurable activity against soluble xylan, a first in LPMOs and an activity of possible biotechnological interest. CvAA9A was selected for study as it has a different pattern of residues known to be important for h-bonding to the +2 glucosyl residue when compared to CxAA9A, which the authors thought may give a different substrate/product profile. In this manuscript, CvAA9A is also more extensively characterized in terms of substrate/product profile. The two enzymes are very similar in activity except with regard Xyl6, which LsAA9A oxidatively cleaves at ~ 1/100 the rate of Cell6. The EPR spectrum of LsAA9A bound to Xyl6 is significantly different to that of Cell6; in contrast CvAA9A suggests Xyl6 does not bind. This indicates a different electronic environment, and this is borne out by the crystal structures of the complexes of LsAA9A-Xyl5 and LsAA9A-Cell5, where Xyl does not directly ligate the copper, and chloride (as superoxide surrogate) binds axially rather than equatorially. The authors see an uncoupling of the rate of cleavage to reductant in the LsAA9A-xylan reaction, and suggest this is due to the superoxide being displaced by water and then attacking the xylan in a “free” state. Although I agree that initial Cu-reduction is no longer rate-limiting, I disagree that this equates to a model where superoxide must be displaced from the copper to react with the Xyl6. I think it much more likely that it remains activated by the copper, and the rate is limited by attaining a productive reaction conformation/configuration with the non-biological substrate. Is there any mass spectrometry evidence that LsAA9A is more susceptible to oxidative damage with Xyl6 as substrate? This might indirectly support the authors’ contention that superoxide must leave the copper to react with Xyl6, although the longer time frame required to productively cleave Xyl6 may lead to ROS damage to the enzyme anyway, as Cu(II)-superoxide will not be stable for long periods of time. However, this is a minor point of interpretation within an excellent manuscript in all other ways.

- We thank this reviewer for this helpful comment and agree that we cannot exclude the possibility of a copper-bound oxygen species acting as the principal oxidant in the cleavage of xylan. We therefore propose the following change to the relevant part of the discussion:

From:

This suggests a different oxidative mechanism may well be in operation for the cleavage of xylose-based substrates by LsAA9A. Indeed, as has already been proposed by Kjaergaard et al⁴¹, activation of O₂ by an AA9 from *Thermoascus aurantiacus* probably gives formation of hydrosuperoxide (HO₂) through associative displacement of a superoxide anion by a water molecule through the axial coordination site on the copper ion. The superoxide could then itself cleave a saccharidic chain by direct attack. Such a mechanism is expected when the axial water molecule on the copper ion is not displaced by the binding of substrate, as is the case with the binding of Xyl5 to LsAA9A. From the low dose LsAA9A:Xyl5 structure described herein, the axial ligand is clearly present on the copper ion, though it is best modelled as a mixture of chloride and water, and the Tyr-O distance (2.86 Å) is not shortened compared to the un-complexed low dose structure (2.72Å - PDB 5ACG). This is in contrast to the low dose LsAA9A:Cell3 structure where the Tyr-O distance is 2.47 Å (PDB 5ACF). Furthermore, the equatorial position in the low dose LsAA9A:Xyl5 is occupied by a water molecule, not a chloride ion, as corroborated by the EPR spectroscopy. Thus, a mechanism by which a ‘free’ superoxide is generated next to the substrate is possible within the LsAA9A-Xyl516 complex. Such a mechanism may be expected to be rate-independent on the redox potential of the reducing agent, since the

rate-limiting step is likely to be hydrogen atom abstraction by the superoxide from the substrate rather than reductive cleavage of the O-O bond. Therefore, the fact that the rate of cleavage of xylan and Xyl6 by LsAA9A is less dependent on reducing agent while the cleavage of the other substrates is strongly dependent (Figure 4) illustrates that a different oxidative mechanism is in operation. Thus the extent of activity on certain substrates is a function of the oxidative species which can be formed at the copper ion which is—in turn—dependent on the substrate.

To:

This suggests a different oxidative mechanism may well be in operation for the cleavage of xylose-based substrates by LsAA9A. Indeed, as has already been proposed by Kjaergaard et al⁴¹, activation of O₂ by an AA9 from *Thermoascus aurantiacus* probably gives formation of a copper-bound superoxide or a free hydroperoxide (HO₂) through associative displacement of a superoxide anion by a water molecule through the axial coordination site on the copper ion. In particular, a superoxide ion bound to the copper in the axial position would be in position to cleave a saccharidic chain by direct attack. Such a mechanism is expected when the axial water molecule on the copper ion is not displaced by the binding of substrate, as is the case with the binding of Xyl5 to LsAA9A. From the low dose LsAA9A:Xyl5 structure described herein, the axial ligand is clearly present on the copper ion, though it is best modelled as a mixture of chloride and water, and the Tyr-O distance (2.86 Å) is not shortened compared to the un-complexed low dose structure (2.72 Å - PDB 5ACG). This is in contrast to the low dose LsAA9A:Cell3 structure where the Tyr-O distance is 2.47 Å (PDB 5ACF). Furthermore, the equatorial position in the low dose LsAA9A:Xyl5 is occupied by a water molecule, not a chloride ion, as corroborated by the EPR spectroscopy. Thus, a mechanism by which a copper-bound superoxide is generated next to the substrate is possible within the LsAA9A-Xyl516 complex. Such a mechanism may be expected to be rate-independent on the redox potential of the reducing agent, since the rate-limiting step is likely to be hydrogen atom abstraction by the superoxide from the substrate rather than reductive cleavage of the O-O bond. Therefore, the fact that the rate of cleavage of xylan and Xyl6 by LsAA9A is less dependent on reducing agent while the cleavage of the other substrates is strongly dependent (Figure 4) illustrates that a different oxidative mechanism is in operation. Thus the extent of activity on certain substrates is a function of the oxidative species which can be formed at the copper ion which is—in turn—dependent on the substrate.

➤

Overall, this manuscript adds significantly to the LPMO literature, and will be of broad interest not only to those interested in plant biomass degradation and LPMOs specifically, but to biochemists and chemists interested in copper enzymes, dioxygen activation, substrate specificity, and polysaccharide chemistry.

All my other comments are typographical or minor in nature.

➤ Typographical changes were made. Minor comments are below:

(1) Line 213. I would refer to SI Tables S4-5 as well as Fig. 5 at this point.

➤ We have added this Table reference as suggested.

(2) SI Fig. 1. Tyr169/164 is indicated as part of the His-brace in the legend, i.e. highlighted in yellow in the figure. Although a (sometimes) Cu ligand, it is generally not included in the definition of the His-brace. Either change the legend to state that yellow indicates Cu ligands rather than His-brace, or remove the yellow coloring from Tyr169/164.

- We changed the legend to indicate that yellow indicates copper ligands.

(3) Line 401. I do not like the terms “on-pathway” and “off-pathway”; for xylan the pathway is not “off-pathway” for the mechanism of xylan cleavage, it is a different pathway. I would prefer “canonical pathway” and “non-canonical pathway” for polysaccharide cleavage.

- We agree with this reviewer’s helpful suggestion and have changed the text accordingly to canonical and non-canonical pathway.

(6) Line 548-550. It would be useful here to indicate the sequence identity between CvAA9A and LaAA9A.

- Done- 41%.

(7) Line 561. In SI Table S5, the CC_{1/2} is not above 50% for LsAA9A (40%). These data should be cut back.

- The manuscript text has been corrected from “above 50%” to “around 50%” not to create confusion for the reader. In fact the CC_{1/2} was also in this case above 50% in the XDS statistics, as stated, but the statistics reported are from the subsequent and last step in XSCALE. At any rate, the 50% cut-off is given to indicate an approximate cut-off in data processing but is dependent on the choice of resolution shell and is very much on the 'safe side' to err on the side of caution, since CC_{1/2} is a relatively new statistic in crystallography. According to data processing in XDS and XSCALE the correlation was significant at the chosen cut-off level for the LsAA9A:Cell₅ data set and thus there is no compelling reason to cut the data back.

(8) Line 599-600. There is no SI Table S2. I think the authors mean Table 2.

- The reviewer is correct, this should read Table 2. We have corrected this.

(10) Fig 1a and Fig. S12o. I found it difficult to distinguish orange and red in these figure panels.

- We have changed the red to purple in figure 1a, and amended the legend, to better distinguish these colours.

(11) Fig 5a, 7a,b, Fig. S11b. What type of electron density map is shown, and what is the contour level? Please add this information to legends.

➤ The information has been added to the legends

(12) Fig. 6a, 7a. The electron density map mesh is too dense to make out the model. Either decrease the width of the mesh lines, or reduce the mesh density.

➤ The mesh has been made lighter as suggested.

Reviewer #3 (Remarks to the Author):

In the paper by Simmons et al., entitled: "Structural and electronic determinants of lytic polysaccharide monooxygenase reactivity on polysaccharide substrates" the authors describe biochemical, biophysical and structural characterizations of two closely related Lytic polysaccharide monooxygenases, LsAA9A from *Lentinus similis* and CvAA9A *Collariella virescens*.

The biochemical activity studies of the two LPMOs show that these cleave a range of different polysaccharides, including cellulose, xyloglucan, mixed-linkage glucan, and glucomannan. One of the two enzymes, LsAA9A, additionally cleaves isolated xylan substrates, this is the first LPMO characterized that show such activity.

The study also includes structural determination of the two studied LPMOs, including an impressive amount of protein ligand complex structures of LsAA9A. These ligand complexes of LsAA9A give detailed insights into the determinants of specificity by this LPMO. The ligand complex structures in combination with electron paramagnetic resonance characterizations of the two enzymes further reveal differences in copper co-ordination upon the binding of xylan compared to glucans. The results presented show that AA9 LPMOs can display different apparent substrate specificities dependent upon substrate type and the electronic state at the copper active site. It is worth noting that this is the second study ever that presents structures of a LPMO with a substrate molecule bound at the catalytic center of the enzyme. The previous study presenting LPMO ligand complex structures was presented partly by the same authors as in the current study.

Overall the study is very novel and includes an impressive amount of work and new results. The different types of studies included in the study are all well planned and well performed. The results from these studies are all of high standard and well-motivated to be included in the paper.

Major comments

The only major criticism of the results presented in the study is the part presenting the results from the phylogenetic analyses of 444 LPMO sequences. This part is neither very informative nor very novel. There are several previous papers presenting much more detailed phylogenetic analyses of LPMOs, including a study by some of the authors of this study. The authors should therefore strongly consider to either expanding the phylogenetic analyses part so that the results presented gives substantial new information compared with previous studies, or remove this part from the paper and instead include the results in the supplementary material section.

- As noted by the reviewer, there are more extensive studies in the literature, and so there seems little justification for expanding this study here. We think however that it is helpful to include a short description for the method and basis for choosing the additional LPMO for study, and the phylogenetic tree is helpful in this regard. It helps to illustrate also the point about prediction of activity based on phylogenetic relationship. As this is one panel of the figure it would not save substantial space to move it to supplementary and, after further consideration, we prefer to keep it within the main text.

Minor comments

The author does use the definition binding sub sites at several places in the paper, e.g- subsite -4 to + 2, row 98, page 5, without giving the reader a good definition for this definition. For most readers, familiar with ligand complex structures of glycoside hydrolases this is a well-known way of defining how a ligand is bound in the substrate binding cleft/tunnel/pocket, but outside the GH research area this nomenclature is not as well-known and need a short clarification early on in the paper.

- This is a helpful comment and we have added this description and a reference.

There are several abbreviations used in the text that should be better described before they are used. Examples of such are: Cell6-2AB, HPAEC, TFA hydrolysates, SOMO, PACE

- We have defined the abbreviations as first use.

REVIEWERS' COMMENTS:

Reviewer #1 (Remarks to the Author):

In my opinion, the authors have satisfactorily addressed the concerns of the reviewers and the manuscript is ready for publication.

Reviewer #2 (Remarks to the Author):

The authors have been very responsive to my minor concerns, and I am entirely satisfied with the subsequent changes to the manuscript. This is a beautiful manuscript combining a large amount of complementary data to present substantive new conclusions on an important class of enzymes involved in biomass degradation.

Reviewer #3 (Remarks to the Author):

The authors have made a good job making the clarifications or corrections suggested/requested by the referees. No further changes are requested at this stage.

Response to reviewers' comments on Simmons et al. Revision 1.

- We thank the referees for their helpful suggestions on the initial manuscript. There are no further changes suggested by the referees on the revised version.

Reviewer #1 (Remarks to the Author):

In my opinion, the authors have satisfactorily addressed the concerns of the reviewers and the manuscript is ready for publication.

Reviewer #2 (Remarks to the Author):

The authors have been very responsive to my minor concerns, and I am entirely satisfied with the subsequent changes to the manuscript. This is a beautiful manuscript combining a large amount of complementary data to present substantive new conclusions on an important class of enzymes involved in biomass degradation.

Reviewer #3 (Remarks to the Author):

The authors have made a good job making the clarifications or corrections suggested/requested by the referees. No further changes are requested at this stage.